# Molecular Interactions between Dietary Lipids and Bone Tissue during Aging

**DOI:** 10.3390/ijms22126473

**Published:** 2021-06-17

**Authors:** Jose M. Romero-Márquez, Alfonso Varela-López, María D. Navarro-Hortal, Alberto Badillo-Carrasco, Tamara Y. Forbes-Hernández, Francesca Giampieri, Irma Domínguez, Lorena Madrigal, Maurizio Battino, José L. Quiles

**Affiliations:** 1Department of Physiology, Institute of Nutrition and Food Technology ‘‘José Mataix”, Biomedical Research Centre, University of Granada, Armilla, Avda. del Conocimiento s.n., 18100 Armilla, Spain; romeromarquez@ugr.es (J.M.R.-M.); alvarela@ugr.es (A.V.-L.); mdnavarro@ugr.es (M.D.N.-H.); albertobadillo@correo.ugr.es (A.B.-C.); 2Nutrition and Food Science Group, Department of Analytical and Food Chemistry, CITACA, CACTI, University of Vigo, 36310 Vigo, Spain; tforbes@uvigo.es; 3Department of Clinical Sicences, Università Politecnica delle Marche, 60131 Ancona, Italy; f.giampieri@univpm.it (F.G.); m.a.battino@univpm.it (M.B.); 4Department of Biochemistry, Faculty of Sciences, King Abdulaziz University, Jeddah 21589, Saudi Arabia; 5Research Group on Foods, Nutritional Biochemistry and Health, Universidad Europea del Atlántico, Isabel Torres 21, 39011 Santander, Spain; Irma.dominguez@uneatlantico.es; 6Universidad Internacional Iberoamericana, Calle 15 Num. 36, Entre 10 y 12 IMI III, Campeche 24560, Mexico; lorena.madrigal@uneatlantico.es; 7International Research Center for Food Nutrition and Safety, Jiangsu University, Zhenjiang 212013, China

**Keywords:** dietary lipids, hallmarks of aging, oxidative stress, saturated fat diet

## Abstract

Age-related bone disorders such as osteoporosis or osteoarthritis are a major public health problem due to the functional disability for millions of people worldwide. Furthermore, fractures are associated with a higher degree of morbidity and mortality in the long term, which generates greater financial and health costs. As the world population becomes older, the incidence of this type of disease increases and this effect seems notably greater in those countries that present a more westernized lifestyle. Thus, increased efforts are directed toward reducing risks that need to focus not only on the prevention of bone diseases, but also on the treatment of persons already afflicted. Evidence is accumulating that dietary lipids play an important role in bone health which results relevant to develop effective interventions for prevent bone diseases or alterations, especially in the elderly segment of the population. This review focuses on evidence about the effects of dietary lipids on bone health and describes possible mechanisms to explain how lipids act on bone metabolism during aging. Little work, however, has been accomplished in humans, so this is a challenge for future research.

## 1. Introduction

Musculoskeletal system is one of the most affected during aging [1]. Age-related bone diseases such as osteoporosis or osteoarthritis are a major public health problem due to the functional disability they generate. In this context, fractures of the hip, spine and wrist have a huge economic and health impact. Furthermore, fractures are associated with a higher degree of morbidity and mortality in the long term, which generates greater financial and health costs [2]. The incidence of fractures is variable between populations. Notwithstanding, this effect will be notably greater in those countries that present a more westernized lifestyle [3]. However, as the world population becomes older, the incidence of this type of disease will increase. Diet and hence nutrition are shown to be useful and modifiable tools for the prevention and management of metabolic bone disorders. The present review focus on the main events related to aging and their impact on bone health. Evidence on the effects of dietary lipids on bone health and the associated molecular mechanisms are analyzed to explain how these nutrients interact between bone metabolism and aging. With this objective, an initial search was performed in PubMed for studies evaluating the effect of different nutritional interventions on bone health. Those interventions modifying dietary fat were identify and then, an individual search for research on the role of any of the identified interventions in bone health was carried out.

## 2. Bone Biology

### 2.1. Bone Structure

Bone is a mineralized connective tissue containing four different types of cells, namely osteoblasts, osteoclasts, bone-lining cells and osteocytes. Osteoblasts are cuboidal cells located along the bone surface responsible for bone formation that usually comprise between 4–6% of the total bone cells. Osteoclasts are multinucleated cells located on the bone surface, isolated or gathered in small groups, responsible for bone resorption [4,5]. Bone-lining cells are flat quiescent osteoblasts that cover bone surfaces avoiding the interaction between the bone matrix and the osteoclasts [6]. Lastly, osteocytes, the most abundant cells in bone matrix, are cells with dendritic morphology located inside lagoons surrounded by a mineralized bone matrix, although its morphology differs according to the type of bone. In this way, osteocytes are interconnected to other neighboring osteocytes, but also to osteoblasts and bone-lining cells on the bone surface, by gap junctions. This organization constitutes the so-called lacuno-canalicular system (LCS) that connect all cytoplasmic processes of the implicated cells and facilitates the intercellular transport of signaling molecules contributing to osteoblast and osteoclast activity regulation [7]. On the other hand, bone extracellular component is composed of inorganic salts and organic matrix that contains collagen proteins (mainly type I collagen), but also non-collagen proteins including proteoglycans and others as osteocalcin, osteopontin, osteonectin, fibronectin and bone morphogenetic proteins (BMP) as well as growth factors and serum proteins [4,5,6].

### 2.2. Bone Remodeling Processes

Bone is a highly dynamic system that is in continuous renovation by means of a process called bone remodeling by which the old bone is replaced by new bone, in a cycle with three steps: initiation of bone resorption by osteoclasts, reversal period and formation of bone matrix by osteoblasts. Bone resorption is performed by osteoclasts that release enzymes such as tartrate-resistant acid phosphatase (TRAP) and H^+^ degrading organic matrix components and acidifying extracellular medium that contributes to hydroxyapatite crystal dissolution. Then, bone matrix synthesis is carried out by osteoblasts that firstly secrete proteins to form the organic matrix and then lead newly formed organic matrix mineralization. At the end of this phase, mature osteoblasts can be converted into bone-lining cells or osteocytes, or directly undergo apoptosis [1,2,3].

### 2.3. Regulation of Bone Remodeling

Bone remodeling is necessary for maintenance and adaptation of the skeleton to mechanical use, fracture healing as well as calcium homeostasis [4]. Importantly, bone mass and bone morphology would be the result of the balance between bone formation and resorption at different areas of the skeleton. This balance depends on the action of several local and systemic factors including multiple cytokines and hormones as well as some dietary components based on bone cell sensitivities to signaling conveyed though such components [8].

Osteoclastogenesis (the formation of osteoclasts) starts with bone marrow-derived mononuclear cells proliferation and differentiation to circulating pre-osteoclasts, an event that requires the expression of the macrophage stimulating factor (M-CSF). Once in the blood and in the presence of receptor activator of nuclear factor κB ligand (RANKL), the pre-osteoclasts fuse with each other forming an immature osteoblast, whose differentiation to mature osteoclast, as well as osteoclast survival, are carried out only in continuous presence of RANKL that induces the expression of multiple gene. In turn, osteoprotegerin (OPG), that is produced by osteoblasts and bone-marrow stromal cells (BMSCs) binds RANKL, preventing the RANKL interaction with its receptor activator of nuclear factor κB (RANK) in osteoclast precursor cells and osteoclasts, reducing osteoclastogenesis and osteoclast lifespan. In turn, several inflammatory mediators including prostaglandin E2, IL-1β, IL-6, IL-11, IL-17 and Tumor Necrosis Factor (TNF)-α have shown to act as osteoclastogenic factors by inducing RANKL and suppressing OPG on calvarial cells [9,10].

On the other hand, osteoblasts that derived from bone marrow mesenchymal stem cells (BMCs) requires activation and overexpression of several transcription factors such as runt-related transcription factor-2 (RUNX2). The newly formed preosteoblasts, that are recognized by the synthesis of type I collagen and bone sialoprotein (BSP), form mature osteoblasts through the expression of RUNX2, osterix (OSX) and several components of the Wnt signaling pathway [11].

## 3. Aging at the Bone

As stated before, during aging one of the most affected systems is the musculoskeletal system [1]. For improving the understanding of age-related bone disorders, the following section will address the main events related to aging and their impact on bone health and metabolism.

### 3.1. Age-Related Changes in Bone Structure

During aging, there is a decline of bone mineral density (BMD) attributed to the loss of trabecular and cortical bone [12]. In humans, and in most of the studied animals, the loss of bone mass is associated with a decrease in the remodeling rate at the trabecular bone compartment but an increase in remodeling rate at the cortical compartment that results in an increase in cortical porosity [13,14]. Consequently, bones become stiffer, and their cross-sectional area decreases, increasing the risk of suffering fractures.

Other predominant feature of age-related bone loss is the accumulation of bone marrow fat [15] as revealed bone biopsies in animals and humans [16,17,18]. This seems to be the consequence of the age-associated changes in growth factors levels and activity of lineage-specific transcription factors that are involved in the differentiation of mesenchymal stem cells (MSCs) into osteoblasts. The main lineage-specific transcription factor that drive MSCs differentiation are RUNX2 for osteoblastogenesis and PPARγ2 for adipogenesis [15,19]. With aging, there is a predominant expression of PPARγ2 by MSCs with a concurrent decrease in RUNX2 expression and, therefore, lower levels of osteoblast differentiation [20]. Such changes would facilitate the differentiation of MSC into adipocytes at the expense of osteoblasts in bone marrow [21,22]. Furthermore, there is an inverse relationship between bone marrow fat volume and bone volume that is independent of sex and that is correlated with the changes observed in people with osteoporosis [23].

### 3.2. Age-Related Changes in Bone Remodeling Processes

Few studies have examined whether there are age-associated changes in the differentiation processes of bone cells or their progenitors. In these context, human studies shown a decrease in serum levels of bone formation markers such osteocalcin, alkaline phosphatase (ALP) and the N-terminal propeptide of type I procollagen (PINP) in both gender during aging [24]. These results are consistent with in vitro and in vivo studies which shown a decrease in the expression of genes encoding for the bone formation markers OPG, ALP and alpha I collagen during aging [25]. In addition, there is a decrease in osteoblast precursor cells lifespan as a consequence of the reduction in the number of stem cells during aging [26].

In contrast, an age-associated increase of urinary bone resorption markers has been reported in humans [24]. In addition, in vitro and studies in animals suggest that there is an increase of gene expression of RANKL and RANKL/OPG ratio during aging [25,27]. The reduction in osteoblasts and the increase of osteoclasts would explain, at least in part, the age-associated decrease in the amount of bone deposited with each remodeling cycle, which results in a decrease in BMD [28].

## 4. Dietary Lipids and Bone Health

Several epidemiological studies suggest that large amounts of fat, especially those containing primarily saturated fatty acids (SFA), may have negative effects on bone health contributing to reduce bone density and increased fracture risk, in older as well as younger people [29,30,31]. That these effects are large enough to increase fracture risk among older individuals is indicated by a study assessing 6250 postmenopausal women [32]. Most of the studies evaluating the role of dietary lipids in bone health using animal models have focused on the effects of high fat diets (HFD) that mostly were rich in saturated fatty acids (SFA-HFD) based in lard or beef tallow (Table 1). Most of them have reported lower values of bone mineral content (BMC) and/or BMD in SFA-HFD-fed animals from 8 to 38 weeks in comparison with those receiving normolipid diets [33,34,35,36,37]. Moreover, these HFDs were able to affect bone health if they were supplied during development [16,38,39,40,41,42,43], but also in adult and aged animals. Regarding bone microarchitecture parameters most of studies described a lower quality in animals fed SFA-HFD compared with those maintained on standard diets [16,33,40,42]. Lastly, structural consequences of consuming HFDs also have been reflected in biomechanical properties of the evaluated bones [16,44]. Likewise, feeding on a HFD for 20 weeks led to lower values of cancellous BMC and bone strength respect than a low-fat diet in adult (40-week-old) rooster fed in such diets respect than a low-fat diet. However, no effects of the amount of dietary fat on mature cortical bone mechanical properties, geometric structure or BMC were observed [45]. Taken together, these reports indicate that HFD have negative effects on bones in rodents at different ages. In addition, these results in confirmed that dietary fat also affect bone health in animals as in humans, indicating that animal models are useful for mechanistic studies regarding the effects of total fat on bone health. Table 1 shows animal studies regarding high-fat diets and bone.

In most of the mentioned studies, it was reported that the SFA-HFD effects on BMD and bone microarchitecture correlated with decreased levels of different circulating bone formation biomarkers. These reduced biomarkers included serum osteocalcin [59,62,63], serum procollagen type 1 amino-terminal propeptide (P1NP) and plasma carboxy-terminal propeptide of type 1 procollagen (P1CP) [40,63]. On other hand, increased levels of the bone resorption biomarkers plasma cross-linked N-telopeptides of bone type I collagen (NTx), urine pyridinoline (Pyr) and deoxypyridinoline (Dpyr) [49,63] were found compared to values observed in standard diet-fed animals.

The mentioned effects on bone remodeling processes are supported by other studies that found increased expression of osteoclast-specific genes [49] in SFA-HFD-fed animals. In turn, serum RANKL levels were higher in the HFD-fed male C57BL/6 mice, although serum OPG levels were not altered [62]. Moreover, a higher number of osteoclasts in trabecular bone area and osteoclast surface in bone surface [40] as well as TRAP activity in serum were higher in SFA-HFD fed animals in comparison with those fed a standard diet suggesting that osteoclastogenesis was enhanced. Likewise, the numbers of colony forming units (CFU)-fibroblastic and CFU-ALP-positive and mineralization nodule in bone marrow stromal cells from male C57BL/6 mice fed a HFD were higher compared with animals fed a standard diet [62]. Likewise, the osteoblast-specific genes including BGLAP2, COL1a1, FGF23 and IGFBP2 were markedly down-regulated [63]. In addition, osteocalcin expression was reduced in bone marrow of HFD-SFA-fed C57BL/6 mice which suggest that these types of diets also reduce bone formation by blocking differentiation of osteoblast progenitor cells [36]. Dietary fats may exacerbate the uncoupling of bone resorption and formation by inhibiting the formation of mature osteoblasts from their stromal progenitor cells, and enhancing adipogenesis [64]. In this sense, histomorphometry of different bones from SFA-HFD-fed animals showed a significant increase in bone marrow adiposity [16,40,41,54,56,65], which seems the result of enhanced BMSCs towards adipocyte [16,40,65] in detriment of osteoblastogenesis [36,37,63].

However, some studies found higher values of BMD in SFA-HFD fed animals in comparison with a low-fat diet and standard diets [50,55]. Likewise, one of the studies reported higher values of trabecular BV/TB and trabecular and cortical thickness (Tb.Th) as well as lower values of Tb.Sp in comparison with standard diet-fed animals. In addition, this study indicates that the prolonged exposure to SFA-HFD decreases bone formation and probably overall bone turnover, which on one hand may protect from bone loss due to aging or estrogen deficiency but on the other hand may decrease bone quality and may predispose to fractures [55]. Lastly, histomorphometric analyses results were in the same sense with lower values of endocortical mineralizing, surface, mineral apposition rate and bone formation rate [55]. In these studies, the SFA-HFD fed animals had higher body weight than their respective controls and could generate greater mechanical load and favoring bone formation.

On the other hand, studies evaluating the role of dietary lipids in bone health from a qualitative standpoint have been also performed in aged male Wistar rats. When isoenergetic and normolipid diets using different fat sources (Virgin olive or sunflower oil) with clear differences on their fatty acid profile (MUFA or n-6 PUFA-rich) have been compared, it has been found that animals lifelong maintained on MUFA-rich diets had higher values of BMD in comparison with those fed n-6 PUFA-rich diets [46]. A similar effect has been observed for the same animals in relation to alveolar bone loss at mandible [47]. In contrast, if PUFA-rich diets are compared with SFA-rich diets, some benefits are also observed for PUFA. In this sense, the administration of supplement containing evening primrose oil (a good source of γ-linolenic acid as well as the n-6 PUFA acid linoleic acid) and fish oil (FO) to elderly women with osteoporosis or osteopenia prevented loss of BMD in the lumbar spine, and BMD in the femur, relative to control women receiving coconut oil, which is rich in SFA [66]. However, in other study where control group receiving no supplemental fat, no effect of a combined evening primrose oil/FO supplement, although it was performed in younger post- and premenopausal women [67]. Importantly, it seems that in PUFA-rich diets, the relative amounts of n-6 and n-3 PUFA, also plays an important role in relation to bone biology. In fact, several studies have shown that animals fed diets rich in n-3 PUFA, usually by adding FO, or with a lower n-6/n-3 ratio had higher BMC and BMD respect than those fed diets with a higher n-6/n-3 ratio [39,44,53,57]. Positive effects of the n-3 PUFA content of the diet also was supported by bone microarchitecture parameters of tibia and femur from growing rats [39]. Moreover, higher activities of serum ALP isoenzymes, including b-ALP, were reported in weanling male Sprague-Dawley rats fed a AIN-93G diet high in n-3 PUFA by adding (70 g/kg) safflower oil and menhaden oil respect than those fed a diet with a lower n-6/n-3 PUFA ratio suggesting that the of n-3 PUFA on bone would be due, at least in part, to these fatty acids (FAs) stimulated bone formation in growing animals [48]. This effect on bone formation is in consistency with other studies reporting a lower bone marrow adiposity in animals fed FO-rich diet in comparison with those fed corn oil-rich diet [57]. Several studies indicate that when the differentiation process is directed toward adipocyte formation, osteoblast formation may be compromised [64,68,69]. Activation of PPAR-γ by FAs, as well as a variety of linoleic acid peroxidation products, can induce adipogenesis and inhibit osteoblastogenesis in vitro [70,71,72]. Therefore, some n-6 PUFA FAs can interact with PPAR-γ to inhibit differentiation of osteoblasts and promote differentiation of adipocytes.

In other animal models, results were in a similar sense but less clear. Piglets receiving a diet supplemented with the n-3 PUFA arachidonic acid (AA) and docosahexanoic acid (DHA) for 14 d had higher weight and greater BMD of the whole body, lumbar spine and femur, although no differences were observed in whole body length, calcium absorption or biochemical markers of bone metabolism [73]. Likewise, newly hatched chicks fed menhaden oil and safflower oil enriched diet (90 g/Kg) had increased fractional labeled trabecular surface and tissue level bone formation rates compared with those fed a soybean oil-enriched diet, although no differences were found for BMC which correlated with the reduced serum ALP activity found [52]. Lastly, in rapidly growing rabbits, feeding on FO-supplemented diet led to reduced tibial structural properties, smaller mid-diaphyseal areal properties and shorter tibiae in comparison with control diet in a pair-fed-fed regimen, but tibial stress at the proportional limit was not significantly affected [74].

Interestingly, dietary supplementation with n-3 PUFA or maintaining a lower n-6/n-3 PUFA ratio in the diet also seem beneficial for bone health when animals were maintained on HFDs. In growing (6 weeks old) male C57BL/6 mice, an HFD for 6 months increased TRAP expression and decreased serum concentrations of osteocalcin and b-ALP. However, if FO was used as dietary fat source, at least in part (3–9%), serum TRAP decreased and higher bone mass was found. Importantly, animals receiving a lower amount of FO had higher femoral BV/TV, Tb.N. Conn.D. and bone mass of second lumbar vertebrae and lower femoral Tb.Sp. [75]. In contrast, no differences in femur BMD or biomechanical strength properties were found in forty-day-old male Sprague–Dawley rats maintained for 65 days on HFDs containing coconut oil, flaxseed oil or safflower oil or a standard diet. Still, those fed high n-3 or high n-6 PUFA diets present stronger femur (as measured by peak load) than those of the standard chow-fed group, after adjustment for significant differences in body weight [51]. This evidenced that n-3 and n-6 PUFA may be beneficial in appropriate amounts, but that diets with high concentrations of FAs may be detrimental during development and in advanced age [48,74,76,77]. More research to determine amounts of individual FAs, ratios among the FAs and interactions with other dietary constituents across the life span is needed before recommendations appropriate to different ages can be made. Figure 1 represents the effects of different diets on bone biology reported in growing and adult animal models.

Despite results found in preclinical studies, only three randomized clinical studies evaluating the effect of nutritional interventions involving dietary fat on bone health were available in PubMed database. Notwithstanding, a search conducted in clinicaltrials.gov database identified a total of nine registered clinical trials on this topic. Main results of the mentioned studies are shown in Table 2. Most of studies addressing the role of HFD or LFD in human bone metabolism was mainly carried out in the context of a hypocaloric diet maintained for a period of one or two years. However, in contrast to animal studies, all these dietary interventions were performed in middle-aged adults and older people. Among these types of interventions, most of the studies found no differences in BMC and BMD as well as in serum bone turnover markers between individuals receiving a HFD and those receiving a standard diet [78,79] or LFD [80]. In contrast, higher BMC and BMD were reported in women consuming a normocaloric LFD (<28%E from fat) respect than those following a standard diet (30%E from fat). Interestingly the last study was initiated in young adults and had a duration of 9 years [81]. The differences between this last one and the previous studies could be also explained because caloric restriction is, by itself, a dietary intervention that has shown a significant reduction of BMC and BMC values in humans with an increase in serum CTX-1 and TRAP and a decrease in b-ALP suggesting that bone resorption was enhanced in detriment of bone formation [82], which could mask the potential role of fat in bone health. Thus, more research evaluating the role of HFD in an isocaloric diet context is necessary.

In the same way, there are 12 additional human studies evaluating dietary lipid role in bone health from a qualitative standpoint. In this context, most of the clinical trials addressing n-3 PUFA supplementation effects on bone biology in older people reported no changes in serum OPG, RANKL, OPG/RANKL ratio, b-ALP, osteocalcin, CTX-1, NTx and calcium levels as well as urinary Pyr, Dpyr and Pyr/Dpyr ratio when they are compared with values obtained in individuals supplemented with n-6 PUFA [83,84], MUFA [85,86] or SFA [87,88]. Indeed, some studies shown a slight reduction of bone turnover markers such NTx and ALP without affect to bone resorption markers (Pyr and Dpyr) between n-3 PUFA supplemented groups and those supplemented with n-6 PUFA [83] or SFA [87]. On the other hand, n-6 PUFA supplementation also did not affect to serum levels of osteocalcin, b-ALP, CTX-1 and calcium as well as urinary NTx, Pyr, Dpyr and calcium excretion levels in comparison with those supplemented with MUFA [89,90] or SFA [83] in a similar age group. Likewise, no differences in BMD, serum calcium, t-ALP, b-ALP, OPG as well as urinary Dpyr and calcium levels were found between extra virgin olive oil (EVOO) (a MUFA-rich fat source) and nut (a n-6 PUFA source) supplementation [91,92,93]. However, a significant post-intervention increases in levels of osteocalcin and P1NP in the supplemented with EVOO, but not in the nut-supplemented one. It should be noted that in most of the reviewed supplementation studies, diet was not controlled beyond supplement, which could increase risk of underestimating the effect of the supplements.

**Table 2 ijms-22-06473-t002:** Studies in humans investigating the role of fat intake on bone tissue.

Population; Age	Intervention vs. Control Diet/Placebo; Duration	Main Changes vs. Control Diet or Placebo	Ref.
42 women and 23 men; 51.3 ± 7.1 y	HFD (61%E fat) vs. SD (30%E fat); 12 m	- No differences in BMC and BMD as well as serum bone crosslaps and urinary Ca excrection levels	[78]
208 women and 99 men; 45.5 + 9.7 y	HFD (>45%E fat) vs. SD (30%E fat); 24 m	- No differences in BMD of spine and hip	[79]
242 women and 182 men; 51.8 ± 8.9 y	HFD (40%E fat) vs. LFD (20%E fat); 24 m	- No differences in BMD of spine, femoral neck and hip	[80]
236 women; 44–50 y	SD (32%E fat) vs. LFD (24%E fat); 18 m	- No differences in BMC of spine and hip as well as BMD of spine ↑ BMD in hip - No differences in serum P1NP and osteocalcin levels	[94]
230 women; 27.3 + 1.1 y	LFD ^dc^ (<28%E fat) + vs. SD ^dc^ (30%E fat); 108 m	↑ BMC and BMD of whole body	[81]
n-3 PUFA supplementation			
3 women and 20 men; 49.3 + 1.6 y	n-3 PUFA rich HFD (37.6%E fat; 6.5%E ALA; n-6/n-3 ratio: 1.6/1) vs. HFD (34.5%E fat; 0.8%E ALA; n-6/n-3 ratio: 9.5/1); 6 wks	↓ serum NTx levels - No differences in serum b-ALP levels	[83]
	n-3 PUFA rich HFD (37.6%E fat; 6.5%E ALA; n-6/n-3 ratio: 1.6/1) vs. n-6 PUFA rich HFD (37.1%E fat; 3.6%E ALA; n-6/n-3 ratio: 3.5/1); 6 wks	- No differences in serum NTx and b-ALP levels	[83]
43 women and 2 men with RA; 57.9 ± 10.8	NCD + n-3 PUFA supplement (2.4 g of n-3 PUFA/d; 1.1 g ALA+ 0.7 g EPA + 0.1 g DPA + 0.4 g DHA) vs. NCD + dairy supplement (2.4 g of SFA/d); 3 m	↓ plasma ALP levels - No differences in urinary Pyr, Dpyr and Pyr/Dpyr ratio	[87]
87 woman and 26 men; 18–67 y	NCD + n-3 PUFA supplement (1.48 g EPA + DHA/d) vs. NCD + placebo (NA g olive oil); 12 wks	- No differences in serum CTX-1 levels	[85]
75 women and 6 men with RA; 49.24 ± 10.46 y	NCD + n-3 PUFA supplement (2.090 g of EPA and 1.165 g of DHA/d) vs. NCD + placebo (NA g of high-oleic-acid sunflower oil); 16 wks	- No differences in serum Ca, b-ALP, osteocalcin and CTX-1 levels	[84]
126 women; 75 ± 7 y	NCD + n-3 PUFA supplement (1.2 g EPA + DHA/d) vs. NCD + placebo (NA g olive oil); 6 m	- No differences in serum b-ALP and osteocalcin	[86]
60 women and 15 men; 35–65 y	NCD + n-3 PUFA/MUFA enriched dairy supplement (23.7 g saturated fat + 5.17 g oleic acid + 0.14 g DHA + 0.20 g EPA/d) vs. NCD + semiskimmed milk (70g saturated fat + 2.05 g oleic acid); 12 m	- No differences in serum OPG, RANKL, OPG/RANKL ratio - No differences in osteocalcin and CTX-1 levels	[88]
n-6 PUFA supplementation			
3 women and 20 men; 49.3 + 1.6 y	n-6 PUFA rich HFD (37.1%E fat; 12.6%E LA; n-6/n-3 ratio: 3.5/1) vs. HFD (34.5%E fat; 7.7%E LA; n-6/n-3 ratio: 9.5/1); 6 wks	- No differences in serum NTx and b-ALP levels	[83]
	n-6 PUFA rich HFD (37.1%E fat; 12.6%E LA; n-6/n-3 ratio: 3.5/1) n-3 PUFA rich HFD (37.6%E fat; 10.5%E LA; n-6/n-3 ratio: 1.6/1); 6 wks	- No differences in serum NTx and b-ALP levels	[83]
38 women and 6 men with RA; 46.2 ± 13.1 y	NCD + n-6 PUFA supplement (2.5 g CLA/d) vs. NCD + placebo (2.5 g of high-oleic-acid sunflower oil); 3 m	↑ osteocalcin and CTX-1 levels - No differences in serum b-ALP levels	[90]
60 men; 49.1+ 6.2 y	NCD + n-6 PUFA supplement (3 g CLA/d) vs. NCD + placebo (NA g palm and bean oil blend/d); 8 wks	- No differences in serum osteocalcin, b-ALP, CTX-1 and Ca as well as urinary NTx, Pyr, Dpyr and Ca levels	[89]
MUFA supplementation			
127 men; 67.9 ± 6.9	CD + EVOO supplementation (>50 mL EVOO/d) vs. CD + nuts supplementation (30 g of mixed walnuts, almonds and hazelnuts); 2 y	↑ osteocalcin and P1NP levels but not with nuts supplementation	[93]
98 women and 104 men; 67.8 ± 6.5	CD + EVOO supplementation (15L EVOO/3 m) vs. CD + nuts supplementation (1350 g of mixed walnuts, almonds and hazelnuts/3 m); 1 y	- No differences in serum Ca, t-ALP, b-ALP, OPG as well as urinary Dpyr and Ca levels	[91]
104 women and 7 men severely obese; 18–40 y	CD + EVOO supplementation (52 mL EVOO/d) vs. CD; 12 wks	- No differences in Pre-post BMD of spine and hip	[92]

Abbreviations: ALA: α-linolenic acid, b-ALP levels: bone-specific alkaline phosphatase, BMC: bone mass content, BMD: bone mass density, Ca: calcium, CD: diet was controlled and proportionate by researchers; CLA: conjugated linoleic acid, CTX-1: C-telopeptide of type I collagen, dc: dietary counseling, DHA: docosahexaenoic acid, DPA: Docosapentaenoic acid, Dpyr: deoxypyridinoline, E: energy, EPA: Eicosapentaenoic acid, EVOO: extra virgin olive oil, HFD: high-fat diet, LA: Linoleic acid, LFD: low fat diet, m: months, MUFA: monounsaturated fatty acid, NA: not available, NCD: CD: diet was not controlled and proportionate by researchers, NTX: N-telopeptides of type I collagen, OPG: Osteoprotegerin, PUFA: Polyunsaturared faty acid, Pyr: pyridinoline, P1NP: procollagen type 1 N-terminal propeptide, RA: rheumatoid arthritis, SD: standard diet, SFA: Saturated fatty acids, t-ALP: total alkaline phosphatase, vs.: versus, wk: weeks, y: years.

## 5. Molecular Mechanisms Operating under the Observed Effects of Dietary Lipids on Bone Health of Interest in Relation to Aging

### 5.1. Mitochondrial Dysfunction and Oxidative Stress

According to the results from some experiments in rodents, the consumption of HFD-SFA diets would lead to increased ROS production at bone in growing (4 weeks old) mice [63]. Several ways for HFD to increase ROS production have been proposed. One is that excessive free fatty acids (FFAs) can enhance the activity of the tricarboxylic acid cycle leading to the increased generation of the reductive equivalent NADH and FADH_2_, which will overload mitochondria and eventually lead to a ROS increase [95]. Another possible explanation lies in the fact of that the excessive deposition of fat contributes to the overclearance of FFAs by mitochondrial β-oxidation, which increases the electron flow of cytochrome c oxidase and the accumulation of ROS [96]. The increase in bone marrow adiposity reported in SFA-HFD-fed animals could also contribute to oxidative stress since FFAs released by the adipocytes contributes to ROS generation and lipid peroxidation as well as to a decrease in superoxide dismutase (SOD) and glutathione peroxidase (GPX), affecting to phosphorylation of Akt, ERK and p38 MAPK [95]. In fact, some studies also found that antioxidants enzymes SOD, catalase (CAT), GPX and glutathione transferase (GST) levels or activity as well as total antioxidant capacity (TAC) at bone were reduced in SFA-HFD-fed groups [63,97,98,99], which is in consistency with decreased expression of NRF1 and NRF2 also reported in SFA-HFD-fed animals [59]. This is supported by the decreased ratio of reduced glutathione to oxidized glutathione (GSH/GSSG) found in femur of male C57BL/6 mice maintained on an SFA-HFD [63].

On the other hand, the impaired bone antioxidant system and the increased generation of ROS would explain the increases levels of oxidative damage markers including malondialdehyde (MDA) found in femur and tibiae [63,97]. In this way, it was also observed a negative correlation between MDA concentrations in the liver and the femoral metaphysic BMD and a positive correlation between SOD, CAT and GPX activity in the liver and the femoral metaphysic BMD in adult female Sprague Dawley rats. Indeed, a strong negative correlation of plasma bone resorption biomarkers NTx with GSH/GSSG ratio and TAC and a strong positive correlation of plasma bone formation biomarkers P1CP with GSH/GSSG ratio and TAC in male C57BL/6 mice have been reported [63]. Oxidative damage markers also were higher in animals fed PUFA-supplemented or -rich diet respect than values found in those receiving other unsaturated fats. For example, supplementation with DHA resulted in oxidative DNA damage in the bone marrow of aged rats, and diets containing large amounts of borage oil (n-6 PUFA-rich) reduced tibia biomechanical properties in aged rats [100]. In a similar way, systemic oxidative damage markers were lower in male Wistar rats lifelong maintained on a MUFA-rich diet compared with those found in rats receiving a n-6 PUFA-rich diet, which correlated with better bone health [46].

Dietary fat-induced oxidative stress may exacerbate the uncoupling of bone resorption and formation by promoting the formation and activation of the osteoclasts necessary to remodel the skeleton and inhibiting the formation of mature osteoblasts from their stromal progenitor cells, and enhancing adipogenesis [64]. Increased intracellular ROS has been shown to stimulate osteoclast differentiation and bone resorption [101,102]. Baek et al. [103] added H_2_O_2_ to human bone marrow monocytes and found that H_2_O_2_ could promote the expression of the RANKL and increase the ratio of RANKL/OPG. This result leads to an increase in the number of osteoclasts, which eventually leads to an increase in bone resorption and a decrease in bone mass. Moreover, antioxidants added to fodder have the effect of eluding this mechanism [63]. Consistent with this idea, Kim et al. found that α-lipoic acid (α-LA), a type of strong antioxidant, suppressed pro-osteoblasts proliferation driven by RANKL and TNF-α (RANKL/TNF-α) independently to restrain activation of NF-κB [103]. Thus, ROS also plays an important role HFD-induced osteoclast differentiation through NF-κB regulation. In contrast, osteoblastic differentiation in mouse and rabbit BMCs [104,105] and calvarial cells [104] has shown to be suppressed under H_2_O_2_-induced oxidative stress. Likewise, mineralization levels and gene expression of the osteogenic markers are diminished in MC3T3-E1 cells incubated with H_2_O_2_ [106]. ROS also act as antagonists of Wnt signaling in osteoblasts through transcription factors FoxO [107,108]. Excessive ROS promote the binding of PPARγ to β-catenin preventing the transcription of β-catenin and downstream TCF/LEF in the Wnt/β-catenin signaling pathway to activate the FoxO transcription factors by elevating the FoxO downstream transcripts. Consequently, osteoblast differentiation and the bone formation would be reduced causing a decrease in the osteoblast number and bone formation. Lastly oxidative damage leads to excessive apoptosis of osteocytes [109,110,111].

Moreover FAs, their oxidation products and oxidized lipoproteins can interact with PPAR-γ to inhibit differentiation of osteoblasts and promote differentiation of adipocytes [112,113]. PPAR-γ is expressed in osteoblasts and activation of PPAR-γ by fatty acids, as well as a variety of linoleic acid peroxidation products, can induce adipogenesis and inhibit osteoblastogenesis in vitro [70,71,72]. In addition, minimally oxidized LDL promoted adipogenic differentiation of murine M2-10B4 marrow stromal cells in the presence of PPAR-γ agonists and inhibited osteoblastic differentiation.

Therefore, oxidative stress could be reflected in RANK-RANKL-OPG system alterations. As said, increased RANKL or decreased OPG local expression can cause bone resorption at various sites of the human skeleton. It has been demonstrated that RANKL is up-regulated, whereas OPG is down-regulated in periodontitis, compared to periodontally health, resulting in an increased RANKL/OPG ratio [114]. In an experiment feeding male rats on diets based on VOO, SO or FO lifelong, circulating levels and gum mRNA amount of RANKL and OPG were measured in the context of alveolar bone resorption [46]. Authors found that RANKL/OPG ratio was higher in old animals compared with the young ones demonstrating that aging is a condition that favors bone loss. Moreover, it was found that VOO and FO, but not SO act in the same way that other studies suggesting that OPG acts as a defensive mechanism during aging in order to avoid an excess of bone destruction induced by an excess of RANKL stimulation. So, although age no doubt affects bone loss, it has been demonstrated that dietary lipids may condition how fast is the bone lost, with the n-6 PUFA being the most deleterious among PUFA.

With aging, resulting from the accumulation of dysfunctional mitochondria and the progressive inefficiency of antioxidant defense mechanisms, ROS production intensifies above normal defense capabilities [115]. Under these conditions, ROS accumulation would favor osteoclastogenesis inhibiting the function of osteoblasts, increasing bone resorption and reducing bone formation during aging [107,108]. The importance of oxidative stress in bone aging has also been suggested by studies in transgenic animal models. In murine genetic models with a global deletion of the genes encoding antioxidant enzymes manifest low bone mass [58,59]. Furthermore, it was observed that the administration of antioxidants prevents bone loss in these animals [58]. Likewise, in older murine models of hyperfunction of NRF2, an elevation of NRF2 was observed to affect the accumulation of bone mass and contribute to bone loss [62,63]. Interestingly, in females, the expression of phase II detoxifying enzymes strictly depends on NRF2 activity, which does not occur in males [64]. These data suggest that there are sex-specific mechanisms to control the defense against ROS in bone.

### 5.2. Apoptosis Dysregulation of Bone Cells

Apoptosis plays a role in the normal maintenance of bone tissue, abnormal bone turnover and aging. More apoptotic formation in osteocytes and osteoblasts were observed in transmission electron microscopy (TEM) images of Vertebrae L4 from SFA-HFD-fed animals in comparison with those obtained in standard diet-fed animals [116]. The excessive apoptosis of osteocytes has been also related to oxidative damage in other studies [110,111,117]. Despite osteocyte-controlled apoptosis may be needed to repair bone microdamage [118], the dysregulation of apoptosis contributes to the imbalance between resorption and bone formation, as well as changes in the mechanical properties of local tissue [119]. The increase of osteocyte apoptosis with age can contribute to bone weakness independent of BMD through, at least, two mechanisms: (1) the formation of areas of micropetrosis due to mineralization of empty lagoons; and (2) interruption of the canalicular system, which reduces microcrack repair [108,120].

Moreover, SFA-HFD has been shown to strongly inhibit osteoclast apoptosis in murine IL-6 deficient models [121] which have been suggested to present cell viability and bone resorptive ability of osteoclasts in mice as consequence of IL-6 depletion [122]. Elevated IL-6-induced inflammation is also shown to be responsible for obesity-associated bone loss in mice [65] suggesting that SFA-HFD effects on bone could be independent on this condition in normal phenotype models, at least in part. In addition, it has been found an increased expression of the apoptosis-related genes bcl-2 and drak1 and more apoptotic cells with age in adult human bone marrow cells (BMCs) [21,27]. The irregularities in apoptosis explain different diseases that result from extensive or inadequate cell death. Consequently, the interruption in bone remodeling, characterized by the survival of osteoclasts and apoptosis of bone forming cells, leads to diseases such as osteoporosis.

### 5.3. Dietary Lipids and Genomic Instability at Bone

Some studies have correlated lower values of femoral BMD and a worst bone microarchitecture parameters with higher values of urinary 8-OHdG, which is a sensitive indicator of oxidative DNA damage [98,123]. Different reports support the idea that DNA damage interferes with normal skeletal maintenance and that the accumulation of damage with aging contributes to bone loss decrease and bone formation since the increased DNA damage would promote senescence and apoptosis processed in osteoblast lineage [124,125,126]. In fact, markers for both, DNA damage and DNA damage response, are increased in osteoprogenitors and osteocytes in old mice and they have been associated with increased cell senescence and apoptosis. The relationship between genomic damage and bone loss is also supported by studies in mice showing that DNA damage due to focal irradiation in bone causes senescence in cells of the osteoblast lineage as well as bone formation decrease and bone loss. Furthermore, irradiation also causes changes in osteoprogenitor cells similar to those observed with aging [126]. The possible implication of genomic instability in bone health alteration has been supported by findings of studies in murine models with DNA repair deficiency that displayed low bone mass associated with low bone formation and increased bone resorption among their multiple symptoms of premature aging [126,127,128,129]. Furthermore, humans with progeroid syndromes caused by deficiencies in DNA repair mechanisms such as xeroderma pigmentosum and trichothiodystrophy [130,131] also present skeletal abnormalities [132,133]. Consumption of SFA-HFDs has been associated with increases in DNA strand breaks in several biological samples including peripheral blood, kidney, liver, pancreas, spleen, brain and bone marrow (Figure 2) [99,134]. In the same sense, a higher number of micronucleated polychromatic erythrocytes has been found in the bone marrow from animals maintained on SFA-HFD [99,134]. Concerning unsaturated fat, aging and DNA damage, Quiles et al. have tested the effects of feeding male Wistar rats with diets containing different fat sources as virgin olive oil (VOO) and sunflower oil (SO) [135]. Lower levels of DNA double-strand breaks in peripheral blood lymphocytes were found in young and old animals fed on VOO. In the same animals it was also analyzed the presence of a particular mitochondrial DNA deletion in the liver [136]. An increase of 6-fold in the deletion was found in the case of old animals fed the VOO-containing diet, meanwhile this increase was 60% in old animals fed the diet containing SO. The lower increase in mtDNA deletion frequency during aging was attributed to the lower number of free radicals produced by VOO. This might also be responsible of the lower alveolar bone resorption found in aged animals fed on VOO compared to those fed on SO [47] and the higher BMD found in the same animals fed on VOO [46], associating then dietary lipids, genetic instability and bone health during aging. Noteworthy, VOO and FO, apart from genomic instability and aging have been reported to live longer than animals fed on SO [137].

### 5.4. Dietary Lipids and Inflammation

Increased expression of circulating inflammatory mediators as well as infiltration of aggravative inflammatory cells in mesenteric adipose tissue [138] and bone marrow adipose tissue [139], have been reported in HFD-fed animals. Some experiments have shown that the chronic low-grade inflammatory circumstances of subjects served a HFD [140] are very similar to that of obese subjects, both of which had an elevated concentration of serum pro-inflammatory factors, with IL-1, IL-6, TNF-α [141,142,143] most notably. Excessive fat intake in the early stages of development can trigger activation of the inflammatory system and cause a significant increase in bone marrow fat content and a decrease in the trabecular BV and BMD [139]. However, the effect of pro-inflammatory factors on bone loss is achieved by affecting different bone cells. Many inflammation cytokines such as IL-6 [121,144] and TNF-α [145] have been identified to activate the RANKL/RANK/ OPG pathway to regulate the bone resorption activity of osteoclasts. It has been observed that TNF-α induces the expression of RANK and IL-1 in the precursors of human bone marrow and mouse osteoclasts [146,147,148]. A HFD can also cause a decrease in OPG levels, which deteriorate the differentiation of osteoblasts and promote the absorption of osteoclasts [149].

On other hand, osteocyte apoptosis can be directly affected by proinflammatory cytokines under inflammatory conditions, which ultimately increases the bone absorption by osteoclasts [150]. On the other hand, TNF-α and IL-1β inhibit osteoblast differentiation and bone formation in mature TNF-α-deficient mice [151]. Furthermore, TNF-α and IL-1β inhibit collagen expression, decreasing bone matrix formation [152]. The expression of RANKL improves in an autocrine way, decreasing the expression and activity of RUNX2 and favoring bone resorption [148,153,154]. These results are confirmed in animal models with genetic deletion of IL-1 where a better conserved bone mass was found but an increase in RANKL [155,156]. Burgeoning experiments have shown that TNF-α can upregulate the expression of sclerotin and Dickkopf-1 (DKK-1) in various animal models [150,157,158], and both can inhibit the Wnt signaling pathway to affect the formation of osteoblasts. Therefore, except for the direct effects of proinflammatory cytokines on osteoclasts and osteocytes, the increase of the inflammatory induced bone formation inhibitor also contributes to a state of low bone formation. These data suggest that inflammation may be responsible, at least in a significant part, for the stimulation of bone resorption during old age.

PGE2, which is derived from arachidonic acid, is thought to contribute to pro-inflammatory processes and high concentrations may inhibit bone formation. Middle-aged (12 months old) male rats, consuming a fat blend containing n-3 PUFA showed lower bone nitric oxide (NO) and bone PGE_2_ production than those found in animals receiving a diet including both n-3 and n-6 PUFA [44]. The n-3 PUFA α-linolenic acid and the n-6 PUFA linoleic acid are converted via a series of desaturation and elongation steps to different FAs, which serve as precursors for the eicosanoids as arachidonic acid. Since n-3 and n-6 fatty acids serve as substrates for the same enzymes along the conversion pathways, it is expected that PGE_2_ production was reduced by lowering the dietary n-6:n-3 ratio [159]. This is supported by the results of other study reporting higher ex vivo PGE_2_ biosynthesis in liver homogenates and bone organ cultures of chicks fed soybean oil compared with the values for those given a combination of menhaden oil and safflower oil. Therefore, ex vivo PGE_2_ production in liver homogenates and bone organ cultures (right femur and tibia) were significantly lower in growing rats fed diets with a lower dietary ratio of n-6: n-3 PUFA than in those fed diets with a higher dietary ratio [48]. Arachidonic acid and DHA addition to the diet led to an increase of the levels of this fatty acid in liver whose levels were positively related to urinary PGE_2_ but negatively related to free linoleic acid in bone in piglets. In addition, an inverse relationship was observed when liver linoleic acid was substituted for liver arachidonic acid as the independent variable [73]. Interestingly, despite a HFD increased the expression of the adipose tissue TNFα in growing C57BL/6 mice, when similar diets containing FO at 3% E such values were lower. However, if the content of FO as increased at 9% no further beneficial effects were found [62].

On the other hand, dietary oils rich in the n-6 fatty acid γ-linoleic acid (GLA), when combined with FO, have been reported to have positive effects on bone health in older subjects [66,67,160,161,162,163,164,165,166]. PGE_2_ production can also be reduced by provision of the n-6 PUFA GLA [162,167] because providing GLA would increase the synthesis of dihommo-γ-linolenic acid but not arachidonic acid, probably due to the limited activity of Δ-5-desaturase [168]. Actually, combined supplementation of GLA and the n-3 PUFA EPA produced similar effects and also prevented the accumulation of arachidonic acid in serum [169]. In addition to reducing synthesis of PGE2, dietary GLA can enhance production of PGE1, which has anti-inflammatory effects [162,165]. Thus, providing GLA in the diet would bypass the initial step converting linoleic acid to GLA and allow synthesis of PGE1 to proceed unimpeded. Therefore, although more work is needed to determine the adequate proportion of FAs, modifying dietary FA profile of the diet to reduce n-6 PUFA amount may prove to be a physiologically effective means to manipulate endogenous prostaglandin synthesis and enhance bone health in advanced age.

### 5.5. Autophagy Alteration and Bone Cell Differentiation

In eukaryotes, autophagy represents a highly evolutionary conserved process, through which macromolecules and cytoplasmic material are degraded into lysosomes and recycled for biosynthetic or energetic purposes. Dysfunction of the autophagic process has been associated with the onset and development of many human chronic pathologies, such as cardiovascular, metabolic and neurodegenerative diseases as well as cancer [170]. It has been observed that the change in osteoblasts towards a non-mineralizing osteocyte phenotype appears to be coordinated by Beclin1-mediated autophagy [171]. A study suggests that the consumption of an SFA-HFD modulates autophagy in osteoblasts directing these cells towards non-mineralizing osteocytes attenuating bone formation. This correlates with histological evaluations of tibiae suggesting that an SFA-HFD promoted terminal differentiation of the osteoblast towards osteocyte in animals. In addition, defective autophagy in osteoblast progenitors (including their descendants) was associated with decreased maturation of osteocytes, as well as retention of endoplasmic reticulum and mitochondria in osteocytes [172]. On the other hand, it has been reported lower levels of LC3B in bone marrow derived primary macrophages and peritoneal macrophages from animals fed SFA-HFD, suggesting a general impairment of autophagy machinery in this cell type, which would affect macrophage polarization contributing to the increased systemic inflammatory state observed in these animals [173]. Therefore, dysregulation of bone metabolism induced by SFA-HFDs also would depend on proteostasis loss in different cell population, directly by regulating autophagy in osteoblasts or indirectly through the specific alteration of macrophage autophagy in cells derived from bone marrow and adipocytes, generating a pro-inflammatory environment that promotes the resorptive processes [9,10,174,175]. The relevance of this process has been revealed in a transgenic DMP-cre mice that mimic various aspects of skeletal aging including a more pronounced bone mass phenotype and lower osteocytes turnover where Atg7, that activates LC3, a central protein that stimulates the autophagy pathway in cells expressing DMP1, was eliminated [172,176]. However, more research is needed regarding the influence of SFA-HFD on the loss of proteostasis to clarify the importance of dietary fat effects on autophagy for bone health maintenance during aging. Table 3. summarizes the potential mechanisms under dietary fats effects on bone health and biology.

### 5.6. Altered Levels of Hormones Involved in Bone Biology

Many hormones are critical to bone turnover and the endocrine system also is affected during aging as most of biological systems, showing changes in its physiological function even during healthy aging. In this sense, age-related changes also can generate a greater sensitivity or a lower capacity of response to different stimuli.

#### 5.6.1. Growth Hormone Axis

Growth hormone (GH) is known to has lipolytic effects and its secretion is inhibited by increases in serum FFAs [177]. However, feeding on an SFA-HFD led to increased serum values of leptin and the insulin-like growth factor (IGF)-1 and lowers plasma values of ghrelin in animals [154,178,179], which could be attributed to the reduced hypothalamic gene expression of growth hormone secretagogue receptor (GHSR) and stomach gene expression of ghrelin found SFA-HFD fed animals [180,181]. In any case, SFA-HFD effect on GH/IGF-1 axis could interfere with bone metabolism directly through leptin metabolism or indirectly through an impairment of glucose metabolism derived from the leptin resistance and the low ghrelin levels, since insulin resistance has been associated with an increased production of proinflammatory cytokines and an alteration of the redox status [182,183]. During aging, IGF-1 content in human bones decreases by 60% [180]. This decrease in the content of serum IGF-1 in the bone matrix is associated with an age-related decrease in BMD and higher risk of hip fractures [184]. Since GH is reduced in the elderly, consumption of HFD may further reduce GH levels thereby increasing osteoporosis risk. This effect could not be fully explained by body fatness, suggesting that the diet may have contributed to the results. In humans, consumption of a single high-fat meal reduced exercise-stimulated GH secretion compared to the GH response obtained after no energy intake or after a high-carbohydrate meal was consumed [185,186]. When humans consumed a HFD for an extended period of time (28 weeks), the FA composition of the diet had differential effects on GH secretion. Specifically, supplementation of a HFD with FO decreased GH levels in young males compared with those consuming a mixed fat (lard, tallow and corn oil) supplement [187]. Likewise, when the effects on animals of diets rich in different PUFAs were compared, it was found that a diet rich in soybean oil modulated pituitary and hypothalamus gene expression of GH, Growth hormone–releasing hormone, growth hormone releasing hormone receptor, gonadotropin-releasing hormone and leptin receptor as well as IGF-1 serum levels but no a diet enriched with FO [188,189]. Moreover, IGFBP-3 levels were higher in the group receiving the FO-rich diet [189]. Interestingly, IGFBP-3 has shown an antiosteoblastogenic activity on bone cells [181,190,191].

#### 5.6.2. Insulin

It is well-documented that aging is characterized by a progressive loss of β-cell function which is associated with a decline of insulin secretion [192,193]. Paradoxically, in most studies, patients with type II diabetes show increased BMD [194]. However, they present alterations in bone microarchitecture including higher cortical porosity and less bone strength, as well as alterations in bone collagen production, which correlates with a reduced biomechanical integrity of the skeleton [194,195,196,197]. Therefore, lifelong feeding on diets that prevent pancreatic alterations and disorders affecting to insulin regulation would result useful for maintaining healthy bones. SFA-HFD fed-animals showed higher plasma levels of insulin and higher area under curve (AUC) in glucose tolerance test [40,42,55,198] and fasting blood glucose [40,55,198]. Glucagon-like peptide (GLP-1), fasting blood glucose has been found to result affected in a similar sense by HFDs in humans [199,200]. These results suggest that SFA-HFD can affect negatively to glucose metabolism through the alteration of metabolic flexibility. In turn, as mentioned above, SFA-HFD-induced insulin resistance and prolongated increases of blood glucose could produce oxidative stress and an increased production of proinflammatory cytokines that could directly or indirectly interfere with bone metabolism.

#### 5.6.3. Thyroid Hormones

It has been observed that SFA-HFD-fed animals had higher serum values of thyroid stimulating hormone (TSH) and lower values of T4 and T3 [201,202,203]. Moreover, immunohistochemical staining showed that animals fed SFA-HFD presented lower levels of thyroid hormone synthesis-related proteins such as TTF-1 and sodium / iodide symporter (NIS) [202]. Similar results were obtained in animals fed SFA-HFD where it was observed a reduced gene expression of thyroid hormone synthesis-related proteins such NIS, thyroglobulin (Tg) and TSH receptor (TSHR) [201]. Nevertheless, there is no direct experimental evidence linking the effect of HFD on bone function through the modification of thyroid function. Despite alterations are highly variable between individuals, in iodine-sufficient areas, in general terms, there is an increase of TSH levels with age [204,205,206]. However, exist controversial results in T4 levels in older people [204,205,206] and only one study shown a decrease of T3 levels [205]. In this sense, exist a high prevalence of subclinical hypothyroidism in the elder population [207]. Notwithstanding, there are studies showing that subclinical hypothyroidism in aged people can be spontaneously corrected over time without any intervention [208,209]. Subclinical hypothyroidism has a controversial influence on bone metabolism in elder people [210,211]. On the other hand, lower serum TSH may be associated with an increased risk of hip fractures [212,213] but not with bone loss [214] in aged population.

On the other hand, a n-6 PUFA rich diet (Safflower oil) did not affect to thyroid hormones in comparison with those fed with SFA rich diet (coconut oil) [178]. In the same way, n-3 PUFA rich diet (FO) did not affect to serum T3 and THS and pituitary gene expression of TSH [189,215]. However, a slightly higher serum T4 was seen in n-3 PUFA rich diet (FO) in comparison with n-6 PUFA-rich (rich in soybean oil) diet-fed animals [179].

#### 5.6.4. Parathyroid Hormone

Parathyroid hormone (PTH) stimulates both anabolic and catabolic process in bone depending on the dose and frequency of the PTH signal [216]. Higher serum PTH levels has been associated with lower BMD [182,183,188,189] and lower serum 1,25(OH)2D3 [190,191] as well as an increase of bone turnover markers [191], which is associated with bone loss [181]. During aging, serum PTH levels increase [217,218,219]. In addition, different factors can cause an increase of serum PTH such as kidney failure [220], estrogen deficiency [215,221], serum 1,25(OH)_2_D_3_ status [222] and age-related drugs intake such loop diuretics [223]. In women, there is some suppression of PTH secretion during the rapid phase of bone loss in the early postmenopausal period. However, in the later stage, PTH secretion gradually increases, which increases bone turnover [224]. This fact is probably due to a better 1,25(OH)_2_D_3_ status and/or a better-preserved renal function [225]. Similarly, PTH secretion also increases in older men [226]. SFA-HFD-fed animals have displayed higher values of serum PTH [227,228,229,230]. The elevated higher serum PTH concentrations in SFA-HFD might contribute to an abnormal regulation of serum 1,25(OH)_2_D concentrations by stimulating 1α-hydroxylase expression and attenuating 24-hydroxylase expression in the kidney [227,228,229]. Higher PTH serum levels were found in elderly men and women after consuming a western diet containing a high content of animal fat [231]. However, some interventions in humans, both young and old, found that a western diet did not affect to 1,25-dihydroxyvitamin D serum levels [231,232]. A SFA-HFD also has been related with a reduced expression of PTH receptor [233]. which could contribute to reduce calcitriol levels despite PTH increases reported in other studies.

On the other hand, higher values of serum PTH, 25-(OH) vitamin D2 and 1,25-(OH)2 vitamin D have been reported in animal receiving a diet enriched with menhaden oil comparison with those fed with a diet enriched with safflower oil [53] suggesting that n-3 PUFA positive effects could be mediated also by this mechanism. Notwithstanding, no differences in serum PTH levels were found between aged male rats (24-months-old) maintained lifelong on diets based on VOO or SO [46]. In adult (40-week-old) roosters, the potential for adverse effects of a HFD on intestinal calcium absorption in the mature animal may be more apparent in cancellous bone, with its faster rate of turnover, than in cortical bone [45]. This is different from the effects of fats on other hormones involved in bone formation. Previous reports showed that parathyroid hormone, for instance, did not change and calcitonin increased when high-fat diets were consumed [34,234].

#### 5.6.5. Sexual Hormones

In women, during menopause, serum levels of 17beta-estradiol decrease by 85–90% and serum estrogen levels decrease by 65–75% of the mean premenopausal levels which contribute to exacerbated bone loss [235,236]. Results of studies on ovariectomized animals suggest that SFA-HFD could aggravated bone loss associated with estrogen deficiency [237,238,239,240], but SFA-HFD diets did not affect to [237] or even increase circulating estradiol concentrations in comparison with animals fed on standard diets [240]. Thus, benefits for different dietary fats do not seem to be mediated by modulating estrogen levels, although this could counteract part of the consequences of their deficiency at advanced age.

## 6. Future Perspectives

Most experimental studies on bone biology and dietary fat have evaluated the consequences of increasing fat amount mainly by using saturated fats as fat source, which usually affect negatively to bone turn-over and structure. On the other hand, the possible role of modifying FA profile has been less studied. PUFA could be beneficial for bone health, but it depends on the control diet and dietary fat amount. The role of n-3 PUFA in bone biology is not clear, probably because this depends on dietary context and relative amount of these FAs. Lastly, MUFA-rich diets also seem beneficial compared with n-6 PUFA-rich diets. Still, there is paucity for studies evaluating the role of dietary fat in bone health at advanced ages, as well as studies comparing bone health-related parameters at different ages to confirm what diets prevent age-associated changes. Moreover, many studies used growing animals, so beneficial effects of some diets could depend on bone formation during this stage instead of bone loss prevention as animals age. Therefore, more studies must be performed to improve diets for preventing the effect of aging on bone health and the usefulness of modify diet composition in the elderly.

## Figures and Tables

**Figure 1 ijms-22-06473-f001:**
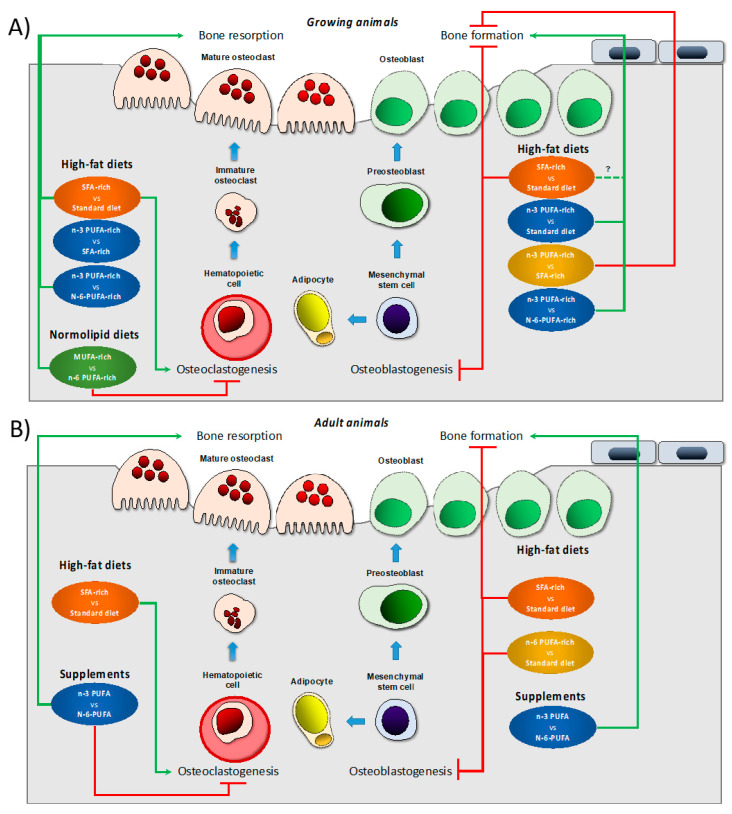
Effect of different dietary interventions concerning dietary fat on bone biology and metabolism reported in growing (**A**) or adult (**B**) animal models. The effects refer changes respect than control diet (indicated after the term vs. in any case) which depend on each study design. Dietary interventions have been subdivided in two categories, high fat diet that, in turn, can be rich in different fat types or supplements with specifical fat types. Green arrows indicate induction in the particular process. Red truncated lines indicate a decrease in the particular process. Abbreviations: MUFA: monounsaturated fatty acids; n-3 PUFA: n-3 polyunsaturated fatty acids; n-6 polyunsaturated fatty acids; SFA: saturated fatty acids, vs.: versus.

**Figure 2 ijms-22-06473-f002:**
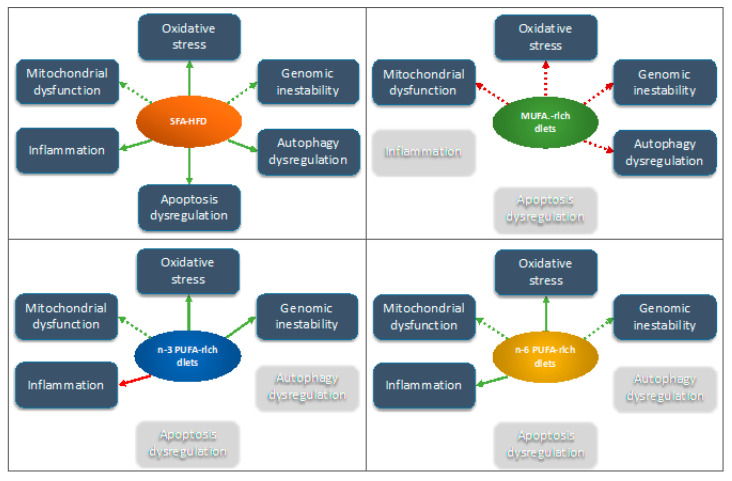
Potential aging-associated mechanisms operating under the observed effects of diets with different dietary fat profile and content on bone biology and metabolism. Green and red colors in the arrows indicate a putative stimulatory or inhibitory effect, respectively. Dashed arrows indicate that to date the effect has been studied only in other tissues different from bone with the exception of systemic inflammation. Abbreviations: HFD: high-fat diet; MUFA: monounsaturated fatty acids, PUFA: polyunsaturated fatty acids, SFA: saturated fatty acids.

**Table 1 ijms-22-06473-t001:** Studies in animal models investigating the role of high-fat diet on bone tissue.

Model; Age	Experimental Diet vs. Control Diet;Duration	Main Changes vs. Control Diet	Ref.
Growing animals			
Male Wistar rat; 3 wks	SFA-HFD ^al^ (40%E beef tallow) vs. SD ^al^ (AIN-93G with soybean oil asa fat source); 8 wks	↓ BMC, BMC/bw, BMD of spine and BV/TV of tibia ↑ Serum t-ALP and b-ALP levels	[38]
	n-6 PUFA-rich HFD ^al^ (40%E soybean oil) vs. SFA-HFD ^al^ (40%E beef tallow); 8 wks	↓ BMD of spine ↓ Serum b-ALP levels	[38]
	n-6 PUFA-rich HFD ^al^ (40%E corn oil) vs. SFA-HFD ^al^ (40%E beef tallow); 8 wks	↓ Serum b-ALP levels	[38]
	n-3 PUFA- rich HFD ^al^ (40%E linseed oil) vs. SFA-HFD ^al^ (40%E beef tallow); 8 wks	↑ BV/TV of tibia	[38]
	n-3 PUFA- rich HFD ^al^ (40%E linseed oil) vs. SD ^al^ (AIN-93G with soybean oil as fat source); 8 wks	↑ Serum b-ALP levels	[38]
Male Wistar rats; Weanling	MUFA-rich SD ^al^ (AIN-93 diet with 9.5%E extra virgin olive oil) vs. n-6 PUFA-rich SD ^al^ (AIN-93 diet 9.5%E sunflower oil); 6 m	↑ Serum osteocalcin	[46]
	MUFA-rich SD ^al^ (AIN-93 diet with 9.5%E extra virgin olive oil) vs. n-6 PUFA-rich SD ^al^ (AIN-93 diet 9.5%E sunflower oil); 6 m	↑ BMD of femur ↑ Serum OPG	[46]
	MUFA-rich SD ^al^ (AIN-93 diet with 9.5%E extra virgin olive oil) vs. n-6 PUFA-rich SD ^al^ (AIN-93 diet 9.5%E sunflower oil); 24 m	↓ Alveolar bone loss of mandibule	[47]
	n-3 PUFA-rich SD ^al^ (AIN-93 diet 9.5%E fish oil); vs. n-6 PUFA-rich SD ^al^ (AIN-93 diet 9.5%E sunflower oil); 24 m	↓ Alveolar bone loss of mandibule	[47]
	MUFA-rich SD ^al^ (AIN-93 diet with 9.5%E extra virgin olive oil) vs. n-3 PUFA-rich SD ^al^ (AIN-93 diet 9.5%E fish oil); 24 m	↓ Alveolar bone loss of mandible	[47]
Male Sprague-Dawley rats; Weanling	HFD ^al^ (AIN93G with added 70 g/kg of safflower oil + menhaden oil) with n-6/n-3 PUFA ratio = 23.8, 9.8 2.6 or 1.2; 42 d	↑ Activities of serum ALP isoenzymes, including b-ALP with lower with n-6/n-3 PUFA ratios	[48]
Female Sprague Dawley rats; 4 wks	HFD ^al^ (12wt% tuna oil) vs. HFD ^al^ (12wt% corn oil); 8 wks	↑ BMD and BMC of tibia ↑ Bone microarchitecture quality of tibia ↑ Serum osteocalcin	[39]
	HFD ^al^ (12wt% flaxseed oil) vs. HFD ^al^ (12wt% corn oil); 8 wks	↑ Bone microarchitecture quality of tibia ↑ Serum osteocalcin	[39]
	HFD ^al^ (12wt% menhaden oil) vs. HFD ^al^ (12wt% corn oil); 8 wks	↑ Bone microarchitecture quality of tibia ↑ Serum osteocalcin	[39]
Male C57BL/6J mice; 5 wks	SFA-HFD ^al^ (60%E lard) vs. SD ^al^ (D12450B chow with 10%E fat); 12 wks	↓ BV/TV of femur and tibia ↓ Bone microarchitecture quality of femur and tibia ↓ Stiffness and maximal load of Femur and tibia ↓ Adipocyte size and adipocyte volume/BV in tibia ↓ Adipogenic formation in isolated MSCs from femoral bone ↓ N.Oc/Tb.Ar and Oc.S/BS of tibia ↓ TRAP-positive osteoclast formation in isolated MSCs from femoral bone	[40]
Male C57BL/6J mice; 6 wks	SFA-HFD ^al^ (60%E lard) vs. SD ^al^ (LabDiet 5LOD with 13.5% E lard); 20 wks	↓ BMC, BV/TV of femur and tibia ↓ Bone microarchitecture quality of femur and tibia ↓ Maximun load, total work, yield load and post yield work of femur ↑ bone marrow adipose tissue volume at epiphysis of tibia and distal tibia	[16]
Male C57BL/6J mice; 6 wks	SFA-HFD ^al^ (60%E lard) vs. SD ^al^ (12%E fat); 12 wks	↑ bone marrow adiposity, adipocyte size and adipocyte no. at femur	[41]
Male C57BL/6 mice; 4 wks	SFA-HFD ^al^ (21.2%E lard) vs. SD ^al^ (4.8%E fat); 13 wks	↓ Plasma levels of P1CP ↑ plasma levels of NTX ↓ Gene expression of osteoblastogenesis specific genes (OCN and COL1a1) and MMP1a in femur and bone marrow ↑ Gene expression of osteoclastogenesis specific genes (NOX2 and RANK) and MMP9 in femur and bone marrow	[49]
Male BALB/cByJ mice; 7 wks	SFA-HFD ^al^ (45%E lard) vs. SD ^al^ (13.5%E fat); 15 wks	↓ BMD, cortical BV/TV, trabecular BV/TV of femur ↓ Bone microarchitecture quality of femur	[42]
Male Wistar rats; 9 wks	SFA-HFD ^al^ (24% fat with 100 g/kg of bw per day of ground nut and 50 g/kg of bw per day dried coconut) vs. SD ^al^; 38 wks	↑ BMD, BMC, cross-sectional area and BV/TV of tibia ↑ Bone microarchitecture of tibia ↑ Serum t-ALP levels	[50]
Male Sprague-Dawley rat; 40 d	SFA-HFD ^al^ (20wt% coconut oil) vs. n-3 PUFA-HFD ^al^ (20wt% flaxseed oil) or n-6 PUFA-HFD ^al^ (20wt% safflower oil); 65 d	- No differences in femur BMD - No differences in biomechanical strength properties ↓ Femur peak load adjusted by bw	[51]
Newly hatched chicks; 4 d	n-3-rich PUFA diet ^al^ (menhaden oil + safflower oil at 90 g/kg) vs. n-6-rich PUFA diet ^al^ (soybean oil + safflower oil at 90 g/kg); 17 d	↑ Fractional labeled trabecular surface ↑ Tissue level bone formation rates ↑ Serum ALP activity	[52]
Adult animals			
Male Sprague-Dawley rat; 200 g	Cholesterol-enriched HFD ^al^ (10.0 g cholesterol, 20.0 g sodium-cholate, and 112.0 g crude fat %per kg dry matter) vs. SD ^al^ (50.83 g crude fat %per kg dry matter); 114 d	↑ Serum b-ALP ↓ Bone calcium loss	[35]
Male F344 × BNF1 rats; 12 m	n-3 PUFA-rich diet (167 g safflower oil + 33 g menhaden oil) vs. n-6 PUFA-rich diet (200 g safflower oil) or n-3 + n-6 PUFA-rich diet (190 menhaden oil + 10 g corn oil); 20 wks	↑ BMC and cortical + subcortical BMD ↑ serum b-ALP activity ↑ serum pyridinoline ↑ urinary Ca	[53]
Male F344 × BNF1 rats; 12 m	n-3 PUFA-rich diet (167 g safflower oil + 33 g menhaden oil) vs. n-6 PUFA-rich diet (200 g safflower oil) or n-3 + n-6 PUFA-rich diet (190 menhaden oil + 10 g corn oil); 20 wks	↑ Peak load, ultimate stiffness and Young’s modulus ↓ Bone formation rate ↓ Osteoclast no. and eroded surface in proximal tibia ↑ Periosteal mineral apposition and formation rates in tibia shaft	[44]
Male C57BL/6J mice; 8 wks	SFA-HFD ^al^ (35wt% lard) vs. SD ^al^ (6wt% fat); 20 wks	↓ Recruitment of progenitor cells to osteoblastic cells ↓ Mineral apposition rate in tibia and vertebrae and bone formation rate tibia ↑ Bone marrow adiposity, adipocyte size and adipocyte no. of proximal tibia ↓ Trabecular BM and cortical thickness ↓ Serum levels of P1NP but not CTX-1 ↓ Percentage of CD73+ and Sca1/CD140a+ cells in MSCs isolated from bone marrow ↓ Short-term proliferation rate and colony-forming units-fibroblast of primary cultures ↑ Gene expression of adipogenic genes (Pparγ2, Lep, Adipoq, Fsp27) b-ALP activity in osteoblast differentiated of isolated MSCs isolated from bone marrow	[54]
Male C57BL/6 mice; 3 m	HFD ^al^ (45%E) vs. SD ^al^ (12%E fat); 11 wks	↑ Trabecular BMD and BV/TV of tibia ↑ Bone microarchitecture quality of tibia ↓ mineral apposition rate and bone formation rate in tibia ↓ Ec.MS/BS in tibia ↑ PmoI and Imax/Cmax in tibia ↓ Serum b-ALP ↑ Serum TRAP activity	[55]
Male C57BL/6J; 4 m	HFD ^al^ (45%E) vs. SD ^al^ (11%E fat); 8 wks	↑ Bone marrow adiposity and adipocyte size of distal femur metaphysis	[56]
Female C57BL/6J mice; 8 m	SFA-HFD ^al^ (45%E lard) vs. SD ^al^ (10%E fat); 8 wks	↓ BMD and BMC of femur	[33]
	MUFA-rich HFD ^al^ (45%E olive oil) vs. SFA-HFD ^al^ (45%E lard); 8 wks	↑ BV/TV of femur ↑ Tb.Th of femur	[33]
Female C57BL/6J mice; 13 m	n-6 PUFA-rich HFD ^al^ (19.5%E corn oil) al vs. SD ^al^ (9.5%E fat); 26 wks	↑ Gene expression of PPARγ at bone marrow adipocytes of femur ↑ Bone marrow adiposity at femur	[57]
	n-3 PUFA-rich HFD ^al^ (19.5%E fish oil) vs. n-6 PUFA-rich HFD ^al^ (19.5%E corn oil); 26 wks	↓ Adipocyte vacuole area of femur ↓ Gene expression of PPARγ at bone marrow adipocytes of femur	[57]
Roosters; 40 wks	HFD ^al^ (8% palmitic acid) vs. LFD ^al^ (8% cellulose); 20 wks	↓ cancellous BMC of femoral condyles and tibial plateau ↓ mechanical properties (bone strength) of the cancellous bone of femoral condyles and tibial plateau - No differences in cortical bone mechanical properties, geometric structure or BMC of tarsometatarsus	[45]
C57BL/6 mice	Cholesterol-enriched HFD ^al^ (15.8wt% fat + 1.25wt% cholesterol) vs. SD ^al^ (6wt% fat), 7 m	↓ BMD and BMC of femur and vertebral BMC ↓ Osteocalcin expression in bone marrow	[36]
C3H/HeJ mice	Cholesterol-enriched HFD ^al^ (15.8wt% fat + 1.25wt% cholesterol) vs. SD ^al^ (6wt% fat), 7 m	- No changes in BMD and BMC	[36]
Aged animals			
OVX female Sprague Dawley; 3 m	DHA-rich diets (HP5 and LP5) High-PUFA diet vs. low-PUFA diet with a ratio of n-6/n-3 PUFAs of 5:1 or 10:1 (110.4 g/kg of fat from safflower oil (110.4 g/kg of high-oleate safflower oil blended with n-3 PUFAs); 12 wks	Fatty acid analyses confirmed that the dietary ratio of 5:1 significantly elevated the amount of DHA in the periosteum, marrow and cortical and trabecular bones of the femur. ↑ BMC and BMD of femur and tibia ↓ Rats fed the LP diets displayed the lowest overall serum pyridinoline and deoxypyridinoline Serum osteocalcin was lowest in the HP groups. Regardless of the dietary PUFA content, DHA in the 5:1 diets (HP5 and LP5) preserved rat femur BMC in the absence of estrogen	[58]
OVX female C57BL/6J mice; 8 wks	AIN-93 diet ^al^ (10%E virgin olive oil) vs. AIN-93 diet ^al^ (10%E refined olive oil); 4 wks	- No differences in BMD, BV/TV and BMC - No differences in Bone microarchitecture	[59]
		Marrow stromal cells from C57BL/6 mice fed a high fat, atherogenic diet failed to undergo osteogenic differentiation in vitro	[37]
Female Wistar rats; 56 m	HFD ^al^ (31%E peanut + canola seed oil); 19 wks	↓ Serum osteocalcin	[59]

Mice were classified as growing (less than 2 months), adults (from 3 to 14 months) or aged (more than 14 months) animals following the recommendation of the Jackson Laboratory [60]. Rats were classified as growing (less than 6 months), adults (from 6 to 18 months) or aged (more than 20 months) according to Sengupta et al. [61]. An exception was made for ovariectomized animals that were considered aged animals since it is an experimental model to replicate the features of human menopause, a major aging-associated change related to bone loss in women. Abbreviations: ^al^: ad libitum, b-ALP levels: bone-specific alkaline phosphatase, BMC: bone mass content, BMD: bone mass density, BS: bone surface, BV: bone volume, BV: Bone volume, bw: body weight, Ca: calcium, CTX-1: C-telopeptide of type I collagen, HFD: high-fat diet, Imax/Cmax: resistance to bending measured across the bone, LFD: low fat diet, m: months, MSCs: mesenchymal stem cells, N.Oc: number of osteoclast, NTX: N-telopeptides of type I collagen, Oc.S: osteoclast surface, OPG: osteoprotegerin, OVX: ovariectomized, P1CP: procollagen type 1 N-terminal propeptide, pMOI: polar moment of inertia, SD: standard diet, SFA: Saturated fatty acids, SFA-HFD: high fat diet rich in saturated fat, t-ALP: total alkaline phosphatase, Tb.Ar: trabecular bone area, Tb.TMD: trabecular tissue mineral density, TRAP: tartrate-resistant acid phosphatase, TV: total volume, vs.: versus, wk: weeks.

**Table 3 ijms-22-06473-t003:** Effects of dietary interventions modifying dietary fats on aging-associated mechanisms operating under the observed effects on bone health.

	Diet	Possible Effect	Ref.
Oxidative stress	SFA-rich HFD	↓ Endogenous antioxidant defense systems ↓ NRF1 and NRF2 gene expression ↑ ROS production ↑ Oxidative damage	[59,63,97,98,99]
	MUFA-rich diet	↓ Oxidative damage	[46,47,136]
	n-6 PUFA-rich diet	↑ Oxidative damage	[46,47,136]
Apoptotis	SFA-rich HFD	↑ osteocyte and osteoblast apoptosis ↓ osteoclast apoptosis	[116,121]
Genomic inestability	SFA-rich HFD	↑ DNA damage and alterations	[99,134]
	n-3 PUFA-rich HFD	↑ DNA damage	[100]
	MUFA-rich diet	↓ DNA damage	[46,136]
	n-6 PUFA-rich diet	↑ DNA damage	[46,136]
Inflammation	SFA-rich HFD	↑ Pro-inflammatory cytokines levels	[62,141,142,143]
	n-3 PUFA-rich HFD	↓ Pro-inflammatory cytokines levels	[62]
Autophagy dysregulation	SFA-rich HFD	↓ Autophagy initiation machinery	[173]
	MUFA-rich diet	↑ Autophagy initiation machinery	[47]

Abbreviations: HFD: high-fat diet, NRF: Nuclear respiratory factor, PUFA: polyunsaturated fatty acid, ROS: reactive oxygen species, SFA: saturated fatty acids.

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
