# Peer review of "Molecular Interactions between Dietary Lipids and Bone Tissue during Aging"

_ijms, 2021, doi:10.3390/ijms22126473_

Round 1
Reviewer 1 Report
The manuscript entitled "Molecular interactions between dietary lipids and bone tissue during aging" is a very interesting article. In this review, the Authors performed a comprehensive analysis concerning evidence about the effects of dietary lipids on bone health and describes possible mechanisms to explain how lipids act on bone metabolism during aging. The review is very informative and overall well-written, the exhaustive overview can provide very useful insight into the field. In my view, the manuscript should be accepted for publication in its present form.
Author Response
The manuscript entitled "Molecular interactions between dietary lipids and bone tissue during aging" is a very interesting article. In this review, the Authors performed a comprehensive analysis concerning evidence about the effects of dietary lipids on bone health and describes possible mechanisms to explain how lipids act on bone metabolism during aging. The review is very informative and overall well-written, the exhaustive overview can provide very useful insight into the field. In my view, the manuscript should be accepted for publication in its present form.
AUTHORS: Authors much appreciate the comments done by the Reviewer.
Reviewer 2 Report
Manuscript title: Molecular interactions between dietary lipids and bone tissue during aging
Manuscript type: review article
This manuscript is a paper describing the correlation between dietary lipids and bone tissue during aging. Bone metabolism and bone changes due to aging are described in detail, and the flow of the paragraph is also followed. Also, the quality of English is sufficient for a thesis to be published. It is considered that this manuscript can be published after undergoing several revisions.
Abstract
In the outline, the goals and contents to be expressed in this study are well described.
main text
The manuscript is written logically and meets the goal of this manuscript. But some modifications are needed.
- Table 1. It is necessary to describe the criteria for dividing growing animals, adult animals and aged animals.
- Please check whether the descriptions of OPG and RANKL in line 190 (reference: 60) are not reversed.
- Contents of line 208: Check that BV/TV is correct, not BV/TB
- The content of line 309: At the end of the sentence “On the other hand” should be rearranged.
- The content of line 454: Please make sure it is VO, not VVO.
- The contents of line 454: Check the position of the abbreviation of FO.
- 5.6.4 PTH is commonly known as a potential anabolic agent. However, the high concentration of PTH in the serum inhibits the activity of osteoblasts and promotes the formation of osteoclasts, as the author said. To avoid confusion among subscribers, please add a little more content in this regard.
- Overall, you should check the abbreviations for oils and hormones.
Author Response
REVIEWER 2
Manuscript title: Molecular interactions between dietary lipids and bone tissue during aging
Manuscript type: review article
This manuscript is a paper describing the correlation between dietary lipids and bone tissue during aging. Bone metabolism and bone changes due to aging are described in detail, and the flow of the paragraph is also followed. Also, the quality of English is sufficient for a thesis to be published. It is considered that this manuscript can be published after undergoing several revisions.
Abstract
In the outline, the goals and contents to be expressed in this study are well described.
main text
The manuscript is written logically and meets the goal of this manuscript. But some modifications are needed.
Table 1. It is necessary to describe the criteria for dividing growing animals, adult animals and aged animals.
AUTHORS: According to Reviewer's recommendations, a brief paragraph has been added to the legend of table 1: “Mice were classified as growing (less than 2 months), adults (from 3 to 14 months) or aged (more than 14 months) animals following the recommendation of the Jackson Laboratory [60]. Rats were classified as growing (less than 6 months), adults (from 6 to 18 months) or aged (more than 20 months) according to Sengupta et al. [61]. An exception was made for ovariectomized animals that were considered aged animals since it is an experimental model to replicate the features of human menopause, a major aging-associated change related to bone loss in women”.
Please check whether the descriptions of OPG and RANKL in line 190 (reference: 60) are not reversed.
AUTHORS: Authors revised the descriptions and were reversed. The abbreviation has been corrected at the appropriate place in the edited manuscript. Line 195 to 196
Contents of line 208: Check that BV/TV is correct, not BV/TB
AUTHORS: authors appreciate the query of the Reviewer. However, BT/TV is correct.
The content of line 309: At the end of the sentence “On the other hand” should be rearranged.
AUTHORS: error has been corrected and is part of edited manuscript.
The content of line 454: Please make sure it is VO, not VVO.
AUTHORS: Authors are grateful to Reviewer for pointing out the mistake. VO abbreviation was wrong, the correct abbreviation is VOO (Virgin olive oil). The abbreviation has been corrected at the appropriate place in the edited manuscript.
The contents of line 454: Check the position of the abbreviation of FO.
AUTHORS: Authors are grateful to Reviewer for pointing out the mistake. The full form has been mentioned at the appropriate place in the edited manuscript.
PTH is commonly known as a potential anabolic agent. However, the high concentration of PTH in the serum inhibits the activity of osteoblasts and promotes the formation of osteoclasts, as the author said. To avoid confusion among subscribers, please add a little more content in this regard.
AUTHORS: Regarding with the advice given by the Reviewer, a clarifying lines have been added to avoid some confusion. Line 688 to 689. “Parathyroid hormone (PTH) stimulates both anabolic and catabolic process in bone depending on the dose and frequency of the PTH signal [220].”
Overall, you should check the abbreviations for oils and hormones.
AUTHORS: Authors are grateful to reviewer for pointing out the mistake. All the full forms have been mentioned at the appropriate place in the edited manuscript.
Reviewer 3 Report
I recommend publication of this review in its current form without reservations.
Author Response
REVIEWER 3
I recommend publication of this review in its current form without reservations.
AUTHORS: Authors much appreciate the comments done by the Reviewer.
Reviewer 4 Report
The review entitled “Molecular interactions between dietary lipids and bone tissue during aging” is an interesting, well-written and well-organized manuscript. The topic is very important.
I have several comments for the authors.
I suggest to add a paragraph reporting the methods used for the literature research (i.e. report the full search strategies for all databases, database used, inclusion and exclusion criteria of the search, etc).
Figure 1 legend should be improved.
Line 152: the authors defined BMD here. However, they used BMD also at line 119. The authors should define abbreviation at first mention. Could the authors check also the other abbreviation (for example SOD, GXP etc)?
In section 4 of the review, the authors focused the attention on dietary studies performed on animals. However, there are few studies on humans. Could the authors add a paragraph describing these studies and clinical trials on humans (checking clinicaltrials.gov) adding also a table?
The authors should add tables and figures also in section 5 to summarize the information.
Author Response
REVIEWER 4
The review entitled “Molecular interactions between dietary lipids and bone tissue during aging” is an interesting, well-written and well-organized manuscript. The topic is very important.
I have several comments for the authors.
I suggest to add a paragraph reporting the methods used for the literature research (i.e. report the full search strategies for all databases, database used, inclusion and exclusion criteria of the search, etc).
AUTHORS: In accordance with the Reviewer's recommendations, a paragraph reporting about the methods used for the literature research was added. Lines 53 to 56: “With this objective, an initial search was performed in PubMed for studies evaluating the effect of different nutritional interventions on bone health. Those interventions modifying dietary fat were identify and then, an individual search for research on the role of any of the identified interventions in bone health was carried out”
Figure 1 legend should be improved.
AUTHORS: the legend of figure 1 has been improved and it is now part of the edited manuscript. Line 267 to 273: “Figure 1. Effect of different dietary interventions concerning dietary fat on bone biology and metabolism reported in growing (A) or adult (B) animal models. The effects refer changes respect than control diet (indicated after the term vs in any case) which depend on each study design. Dietary interventions have been subdivided in two categories, high fat diet that, in turn, can be rich in different fat types or supplements with specifical fat types. Green arrows indicate induction in the particular process. Red truncated lines indicate a decrease in the particular process. Abbreviations: MUFA: monounsaturated fatty acids; n-3 PUFA: n-3 polyunsaturated fatty acids; n-6 polyunsaturated fatty acids; SFA: saturated fatty acids, vs.: versus.”
Line 152: the authors defined BMD here. However, they used BMD also at line 119. The authors should define abbreviation at first mention. Could the authors check also the other abbreviation (for example SOD, GXP etc)?
AUTHORS: Authors are grateful to Reviewer for pointing out the mistake. All the full forms have been mentioned at the appropriate place in the edited manuscript.
In section 4 of the review, the authors focused the attention on dietary studies performed on animals. However, there are few studies on humans. Could the authors add a paragraph describing these studies and clinical trials on humans (checking clinicaltrials.gov) adding also a table?
AUTHORS: In accordance with the Reviewer's recommendations, a paragraph and a table evaluating the role of dietary lipids on bone health in human were added. Lines 297 to 336: “Despite results found in preclinical studies, only three randomized clinical studies evaluating the effect of nutritional interventions involving dietary fat on bone health were available in PubMed database. Notwithstanding, a search conducted in clinicaltrials.gov database identified a total of nine registered clinical trials on this topic. Main results of the mentioned studies are shown in Table 2. Most of studies addressing the role of HFD or LFD in human bone metabolism was mainly carried out in the context of a hypocaloric diet maintained for a period of one or two years. However, in contrast to animal studies, all these dietary interventions were performed in middle-age adults and older people. Among these types of interventions, most of the studies found no differences in BMC and BMD as well as in serum bone turnover markers between individuals receiving a HFD and those receiving a standard diet [80,81] or LFD [82]. In contrast, higher BMC and BMD were reported in women consuming a normocaloric LFD (<28%E from fat) respect than those following a standard diet (30%E from fat). Interestingly the last study was initiated in young adults and had a duration of 9 years [83]. The differences between this last one and the previous studies could be also explained because caloric restriction is, by itself, a dietary intervention that has shown a significant reduction of BMC and BMC values in humans with an increase in serum CTX-1 and TRAP and a decrease in b-ALP suggesting that bone resorption was enhanced in detriment of bone formation [84], which could mask the potential role of fat in bone health. Thus, more research evaluating the role of HFD in an isocaloric diet context is necessary.
In the same way, there are 12 additional human studies evaluating dietary lipid role in bone health from a qualitative standpoint. In this context, most of the clinical trials addressing n-3 PUFA supplementation effects on bone biology in older people reported no changes in serum OPG, RANKL, OPG/RANKL ratio, b-ALP, osteocalcin, CTX-1, NTx and calcium levels as well as urinary Pyr, Dpyr and Pyr/Dpyr ratio when they are compared with values obtained in individuals supplemented with n-6 PUFA [85,86], MUFA [87,88] or SFA [89,90]. Indeed, some studies shown a slight reduction of bone turnover markers such NTx and ALP without affect to bone resorption markers (Pyr and Dpyr) between n-3 PUFA supplemented groups and those supplemented with n-6 PUFA [85] or SFA [89]. On the other hand, n-6 PUFA supplementation also did not affect to serum levels of osteocalcin, b-ALP, CTX-1 and calcium as well as urinary NTx, Pyr, Dpyr and calcium excretion levels in comparison with those supplemented with MUFA [91,92] or SFA [85] in a similar age group. Likewise, no differences in BMD, serum calcium, t-ALP, b-ALP, OPG as well as urinary Dpyr and calcium levels were found between extra virgin olive oil (EVOO) (a MUFA-rich fat source) and nut (a n-6 PUFA source) supplementation [93–95]. However, a significant post-intervention increases in levels of osteocalcin and P1NP in the supplemented with EVOO, but not in the nut-supplemented one. It should be noted that in most of the reviewed supplementation studies, diet was not controlled beyond supplement, which could increase risk of underestimating the effect of the supplements.”
Table 2. Studies in humans investigating the role of fat intake on bone tissue. |
|||
Population; age |
Intervention vs. control diet/placebo; duration |
Main changes vs. control diet or placebo |
Ref. |
42 women and 23 men; 51.3 ± 7.1 y |
HFD (61%E fat) vs SD (30%E fat); 12 m |
- No differences in BMC and BMD as well as serum bone crosslaps and urinary Ca excrection levels
|
[80] |
208 women and 99 men; 45.5 + 9.7 y
|
HFD (>45%E fat) vs SD (30%E fat); 24 m |
- No differences in BMD of spine and hip |
[81] |
242 women and 182 men; 51.8 ± 8.9 y
|
HFD (40%E fat) vs LFD (20%E fat); 24 m |
- No differences in BMD of spine, femoral neck and hip
|
[82] |
236 women; 44-50 y |
SD (32%E fat) vs LFD (24% E fat); 18 m |
- No differences in BMC of spine and hip as well as BMD of spine ↑ BMD in hip - No differences in serum P1NP and osteocalcin levels
|
[96] |
230 women; 27.3 +1.1 y |
LFDdc (<28%E fat) + vs SDdc (30%E fat); 108 m |
↑ BMC and BMD of whole body
|
[83] |
n-3 PUFA supplementation
|
|||
3 women and 20 men; 49.3 + 1.6y |
n-3 PUFA rich HFD (37.6%E fat; 6.5%E ALA; n-6/n-3 ratio: 1.6/1) vs HFD (34.5%E fat; 0.8%E ALA; n-6/n-3 ratio: 9.5/1); 6 wks |
↓ serum NTx levels - No differences in serum b-ALP levels
|
[85] |
|
n-3 PUFA rich HFD (37.6%E fat; 6.5%E ALA; n-6/n-3 ratio: 1.6/1) vs n-6 PUFA rich HFD (37.1%E fat; 3.6%E ALA; n-6/n-3 ratio: 3.5/1); 6 wks
|
- No differences in serum NTx and b-ALP levels
|
[85] |
43 women and 2 men with RA; 57.9 ± 10.8 |
NCD + n-3 PUFA supplement (2.4g of n-3 PUFA/d; 1,1 g ALA+ 0,7 g EPA + 0,1 g DPA + 0,4 g DHA) vs NCD+ dairy supplement (2.4g of SFA/d); 3 m |
↓ plasma ALP levels - No differences in urinary Pyr, Dpyr and Pyr/Dpyr ratio
|
[89] |
87 woman and 26 men; 18–67 y |
NCD + n-3 PUFA supplement (1,48 g EPA+DHA/d) vs NCD + placebo (NA g olive oil); 12 wks |
- No differences in serum CTX-1 levels
|
[87] |
75 women and 6 men with RA; 49.24 ± 10.46 y |
NCD + n-3 PUFA supplement (2.090 g of EPA and 1.165 g of DHA/d) vs NCD + placebo (NA g of high-oleic-acid sunflower oil); 16 wks |
- No differences in serum Ca, b-ALP, osteocalcin and CTX-1 levels
|
[86] |
126 women; 75 ± 7 y |
NCD + n-3 PUFA supplement (1.2 g EPA+DHA/d) vs NCD + placebo (NA g olive oil); 6 m |
- No differences in serum b-ALP and osteocalcin
|
[88] |
60 women and 15 men; 35-65 y |
NCD + n-3 PUFA/MUFA enriched dairy supplement (23.7g saturated fat + 5.17 g oleic acid + 0.14 g DHA + 0.20 g EPA/d) vs NCD + semiskimmed milk (70g saturated fat + 2.05 g oleic acid); 12 m |
- No differences in serum OPG, RANKL, OPG/RANKL ratio - No differences in osteocalcin and CTX-1 levels
|
[90] |
n-6 PUFA supplementation
|
|||
3 women and 20 men; 49.3 + 1.6 y |
n-6 PUFA rich HFD (37.1%E fat; 12.6%E LA; n-6/n-3 ratio: 3.5/1) vs HFD (34.5%E fat; 7.7%E LA; n-6/n-3 ratio: 9.5/1); 6 wks
|
- No differences in serum NTx and b-ALP levels |
[85] |
|
n-6 PUFA rich HFD (37.1%E fat; 12.6%E LA; n-6/n-3 ratio: 3.5/1) n-3 PUFA rich HFD (37.6%E fat; 10.5%E LA; n-6/n-3 ratio: 1.6/1); 6 wks
|
- No differences in serum NTx and b-ALP levels |
[85] |
38 women and 6 men with RA; 46.2 ± 13.1 y |
NCD + n-6 PUFA supplement (2.5 g CLA/d) vs NCD + placebo (2.5 g of high-oleic-acid sunflower oil); 3 m |
↑ osteocalcin and CTX-1 levels -No differences in serum b-ALP levels
|
[92] |
60 men; 49.1+ 6.2 y |
NCD + n-6 PUFA supplement (3 g CLA/d) vs NCD + placebo (NA g palm and bean oil blend/d); 8 wks |
- No differences in serum osteocalcin, b-ALP, CTX-1 and Ca as well as urinary NTx, Pyr, Dpyr and Ca levels
|
[91] |
MUFA supplementation
|
|||
127 men; 67.9 ± 6.9 |
CD + EVOO supplementation (>50mL EVOO/d) vs CD + nuts supplementation (30g of mixed walnuts, almonds and hazelnuts); 2 y
|
↑ osteocalcin and P1NP levels but not with nuts supplementation
|
[95] |
98 women and 104 men; 67.8 ± 6.5
|
CD + EVOO supplementation (15L EVOO/3 m) vs CD + nuts supplementation (1,350g of mixed walnuts, almonds and hazelnuts/3 m); 1 y
|
- No differences in serum Ca, t-ALP, b-ALP, OPG as well as urinary Dpyr and Ca levels
|
[93] |
104 women and 7 men severely obese; 18-40 y |
CD + EVOO supplementation (52mL EVOO/d) vs CD; 12 wks |
- No differences in Pre-post BMD of spine and hip
|
[94] |
Abbreviations: ALA: α-linolenic acid, b-ALP levels: bone-specific alkaline phosphatase, BMC: bone mass content, BMD: bone mass density, Ca: calcium, CD: diet was controlled and proportionate by researchers; CLA: conjugated linoleic acid, CTX‐1: C‐telopeptide of type I collagen, dc: dietary counseling, DHA: docosahexaenoic acid, DPA: Docosapentaenoic acid, Dpyr: deoxypyridinoline, E: energy, EPA: Eicosapentaenoic acid, EVOO: extra virgin olive oil, HFD: high-fat diet, LA: Linoleic acid, LFD: low fat diet, m: months, MUFA: monounsaturated fatty acid, NA: not available, NCD: CD: diet was not controlled and proportionate by researchers, NTX: N-telopeptides of type I collagen, OPG: Osteoprotegerin, PUFA: Polyunsaturared faty acid, Pyr: pyridinoline, P1NP: procollagen type 1 N‐terminal propeptide, RA: rheumatoid arthritis, SD: standard diet, SFA: Saturated fatty acids, t-ALP: total alkaline phosphatase, vs: versus, wk: weeks, y: years. |
The authors should add tables and figures also in section 5 to summarize the information.
AUTHORS: In accordance with the Reviewer's recommendations, a figure and table sumarizing the molecular mechanisms operating under the observed effects of dietary lipids on bone health were added.
Figure 2. Potential aging-associated mechanisms operating under the observed effects of diets with different dietary fat profile and content on bone biology and metabolism. Green and red colors in the arrows indicate a putative stimulatory or inhibitory effect, respectively. Dashed arrows indicate that to date the effect has been studied only in other tissues different from bone with the exception of systemic inflammation. Abbreviations: HFD: high-fat diet; MUFA: monounsaturated fatty acids, PUFA: polyunsaturated fatty acids, SFA: saturated fatty acids.
Table 3. Effects of dietary interventions modifying dietary fats on aging-associated mechanisms operating under the observed effects on bone health.
|
Diet |
Possible effect |
Ref. |
Oxidative stress |
SFA-rich HFD |
↓ Endogenous antioxidant defense systems ↓ NRF1 and NRF2 gene expression ↑ ROS production ↑ Oxidative damage |
[63,99–101,180] |
MUFA-rich diet
|
↓ Oxidative damage |
[46,67,139] |
|
n-6 PUFA-rich diet
|
↑ Oxidative damage |
[46,67,139] |
|
Apoptotis |
SFA-rich HFD
|
↑ osteocyte and osteoblast apoptotis ↓ osteoclast apoptotis |
[118,123] |
Genomic inestability |
SFA-rich HFD
|
↑ DNA damage and alterations |
[136,137] |
n-3 PUFA-rich HFD
|
↑ DNA damage
|
[102] |
|
MUFA-rich diet
|
↓ DNA damage
|
[46,139] |
|
n-6 PUFA-rich diet |
↑ DNA damage
|
[46,139] |
|
Inflammation |
SFA-rich HFD |
↑ Pro-inflammatory cytokines levels
|
[62,144–146] |
n-3 PUFA-rich HFD
|
↓ Pro-inflammatory cytokines levels |
[62] |
|
Autophagy dysregulation |
SFA-rich HFD
|
↓ Autophagy initiation machinery |
[176] |
MUFA-rich diet
|
↑ Autophagy initiation machinery
|
[67] |
|
Abbreviations: HFD: high-fat diet, NRF: Nuclear respiratory factor, PUFA: polyunsaturared faty acid, ROS: reactive oxygen species, SFA: saturated fatty acids |
Round 2
Reviewer 4 Report
The manuscript improved.